# Dehumidification Potential of a Solid Desiccant Based Evaporative Cooling System with an Enthalpy Exchanger Operating in Subtropical and Tropical Climates

**Ramadas Narayanan [1,*], Edward Halawa [1,2] and Sanjeev Jain [3]**

1   School of Engineering and Technology, Central Queensland University, 6 University Drive, Bundaberg, QLD 4670, Australia
2   Future Industries Institute, University of South Australia, Mawson Lakes, SA 5095, Australia
3   Indian Institute of Technology Delhi, Hauz Khas, New Delhi 110 016, India
*   Correspondence: r.narayanan@cqu.edu.au or ramadas.n@gmail.com

**Abstract:** The technical and economic attractiveness of a solid desiccant based evaporative cooling system depend on several factors: Configuration of the system components and their individual performance, availability of cheap but reliable regeneration heat source. In the tropical and subtropical regions, the air conditioning systems are expected to address not only the sensible loads but also, and most importantly—the loads due to higher outside humidity levels that can severely affect the thermal comfort of the building occupants. This paper reports on the dehumidification potentials of solid desiccant based evaporative cooling systems with an enthalpy exchanger operating in subtropical and tropical climates. In particular, the study presents the cooling and dehumidification capabilities of the enthalpy exchanger observed through the impact of its sensible and latent effectiveness on the thermal comfort of the conditioned space. The key performance indicators are split into two groups: (1) the thermal comfort of the conditioned space, and (2) the coefficient of performance. It was found that this cooling system with enthalpy exchanger performed better than the one without enthalpy exchanger in terms of dehumidification; however, the impact depends on the climate where the system operates.

**Keywords:** air conditioning; desiccant wheel; enthalpy exchanger; evaporative cooling; thermal comfort

---

## 1. Introduction

Solid desiccant based evaporative air conditioning system (SDEAC) combines the dehumidifying capability of a solid desiccant system with the cooling capability of an evaporative cooling system to provide thermal comfort to the occupants of a conditioned space. The system's main appeal lies in the fact that, if properly designed, it consumes less electrical energy than the conventional refrigerative air conditioning systems. A recent optimisation study has indicated that an electrical coefficient of performance higher than 20 [1] could be attained. Low electrical energy consumption is made possible using thermal energy sources such as solar energy or waste heat as the regeneration heat source.

Recently, there have been several research works on solid desiccant evaporative air conditioning which investigate various related technical and economic performances of the systems and components [1–14]. Earlier, Henning [15] presented a review on various possible system configurations suitable for certain climatic conditions whilst Camargo and Ebinuma [16] presented thermodynamic equations of the system with focus on the humid climate applications.

To be able to compete technically and economically with the conventional refrigerative systems, SDEAC must be properly designed to achieve the following: (1) to minimise the parasitic energy consumption, (2) to minimise the use of regeneration heat, (3) to be on par with the conventional refrigerative systems in delivering the thermal comfort. To achieve the above goals, several system configurations have been proposed that are expected to suit certain climatic conditions [15]. Other configurations can be found in [5,7,17]. The standard desiccant cooling cycle mentioned in [15] was recently studied to investigate the system performance for the Brisbane climate zone in Australia [18] with mixed results; i.e., a slightly better electrical COP compared with the refrigerative system but with thermal comfort conditions of the conditioned spaces not fully satisfied.

While relatively comprehensive analyses have been carried out to study the performance of the SDEAC [1,6–8], these previous studies have largely overlooked the impact of such a system in terms of delivering the thermal comfort. For instance, the impact of outdoor humidity was mentioned in [8], but the air temperature and humidity level in the conditioned space was not discussed. The same observation is true for a very recent study on two-stage desiccant system [19,20].

The success of SDEAC largely depends of its dehumidification potential. This paper evaluates the dehumidification capability of a SDEAC equipped with an enthalpy exchanger. In the previous study [18], the performance of the single stage SDEAC reported in the earlier studies [5–8] was evaluated. The findings [15] echo the conclusions presented in [5]. However, no research work has been carried out on the performance analysis of SDEAC system for residential applications in tropical and subtropical climates where latent cooling load is an important factor. To develop this system as an alternative to the energy intensive vapor compression system, it is important to assess the dehumidification potential which will help the design of systems for these regions. Therefore, the current study focusses on dehumidification and cooling characteristics of SDEAC system for a residential building in the climate conditions in the Australian cities of Brisbane (27.4698° S, 153.0251° E), Townsville (19.2590° S, 146.8169° E) and Darwin (12.4634° S, 130.8456° E), which represent three distinct subtropical and tropical humid regions, respectively.

The project investigates the effects of adding an enthalpy exchanger in the desiccant evaporative cooling system cycle for subtropical and tropical climates. This study targets an objective method of assessing the thermal performance of the SDEAC system with enthalpy exchanger, i.e., direct evaluation of its capability to the required air temperature and humidity levels in the conditioned space. The direct evaluation of the system capability to deliver acceptable thermal comfort condition is always essential to develop it as a potential alternative to energy intensive refrigerated air-conditioning systems. Therefore, the paper is HVAC application oriented and includes the numerical models of SDEAC system using TRNSYS, which is a well-established software in the HVAC industry with all its components validated with experimental results.

## 2. An Overview of SDEAC System Configurations

A standard SDEAC system consists of a desiccant wheel, an energy recovery wheel, evaporative coolers (at supply and exhaust streams) and a regeneration (reactivation) source [15], as shown in Figure 1. The desiccant wheel dehumidifies the air but increases its temperature significantly (process 1→2). The first cooling of this supply air takes place in the energy recovery wheel (2→3), a constant humidity process. Before being supplied to the conditioned space, the air is further cooled in a direct (or indirect) evaporative cooler (3→4) where the air humidity may be slightly raised but kept well below the comfort requirement. As the supply air enters the conditioned space, its temperature and humidity increase due to the rooms sensible and latent heat gains (4→5).

The air leaving the air-conditioned space is directed to the exhaust where it undergoes several psychrometric processes starting with cooling in an evaporative cooler (5→6). This brings the air close to the saturation line with or without humidification, depending on the evaporative cooling process (direct or indirect). In the energy recovery wheel, the air is preheated (6→7) before it passes through the heater (7→8) where its temperature is further raised so that it will be able to desorb the

water from the desiccant in the regeneration side of the desiccant wheel (8→9). The regeneration heat source (E) can be either a solar thermal system or waste heat. A typical wheel has a 50:50 split between dehumidification and regeneration area.

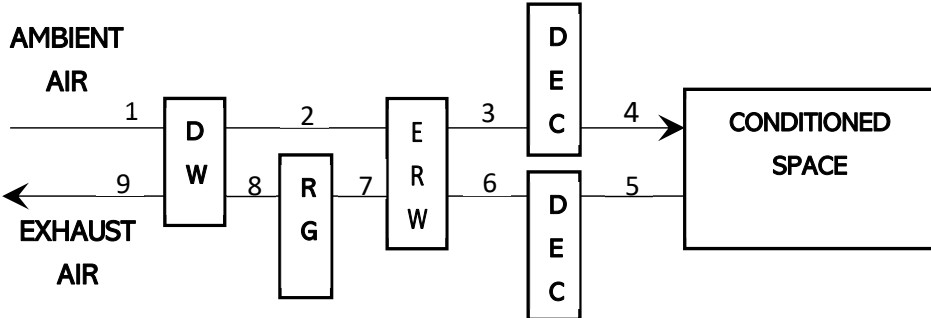

**Figure 1.** Solid desiccant evaporative cooling system. DW: desiccant wheel, ERW: energy recovery wheel, DEC: Direct Evaporative cooler, RG: regeneration source.

According to Henning [15], this configuration is suitable for the temperate climate only. For a more humid climate, the dehumidification capacity of such a system is not adequate to enable the evaporative cooling system to provide the cooling at an acceptable room humidity level. This observation is supported by a recent numerical study for the subtropical climate of Brisbane [18] where for a significantly high number of hours the resulting humidity levels fall beyond the ASHRAE upper thermal comfort boundary. Thus, although according to [6–8] such a system has a relatively high electrical and thermal coefficient of performance in Darwin and Brisbane, the system may not be able to provide the comfortable humidity levels in the conditioned spaces. In the previous studies [6–8] the room humidity level was not discussed.

Typical air temperature and humidity levels at the inlet and outlet of each component of this configuration extracted from [15] are given in Table 1.

**Table 1.** Typical air temperature and humidity levels at the inlet and outlet of each component of the solid desiccant based evaporative air conditioning (SDEAC) system whose configuration is shown in Figure 1. DW: Desiccant wheel, ERW: Energy recovery wheel, DEC: Direct evaporative cooler, RG: Regenerator (The green area represents the process on the supply side whilst the blue area represents the process on the exhaust side with the arrows indicating the flow directions).

| °C | 35 | | 53 | | 27 | | 20 | | | Temperature |
|---|---|---|---|---|---|---|---|---|---|---|
| → | | **DW** | → | ERW | → | DEC | → | | | Supply side |
| g/kg | 11 | | 6 | | 6 | | 8.5 | | | Humidity ratio |
| °C | 48 | | 75 | | 45 | | 20 | | 27 | Temperature |
| ← | | DW | ← | RG | ← | ERW | ← | DEC | ← | Exhaust side |
| g/kg | 19 | | 13 | | 13 | | 13 | | 11 | Humidity ratio |

An alternative to the above configuration is shown in Figure 2 where an enthalpy exchanger (EE) is introduced to pre-cool and pre-dehumidify the ambient air using the return air from the building [15].

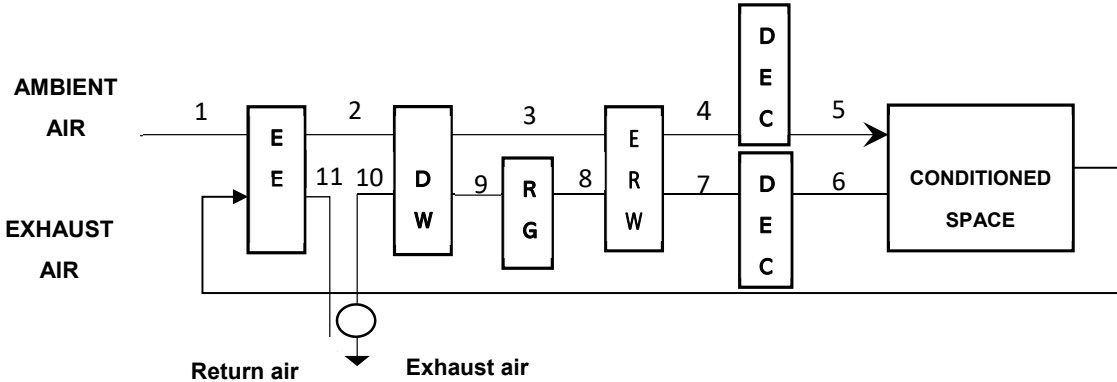

**Figure 2.** Solid desiccant based evaporative cooling system for climate with high humidity. EE: enthalpy exchanger, DW: desiccant wheel, ERW: energy recovery wheel, DEC: Direct Evaporative cooler, RG: regeneration source.

The enthalpy exchanger referred to in this paper is an air to air heat recovery device which allows heat and mass transfer between two air streams passing through the device as shown in Figure 3. The mass transfer is made possible through a permeable membrane separating the two streams [21]. Two of the important parameters of the enthalpy exchanger are the latent and sensible effectiveness.

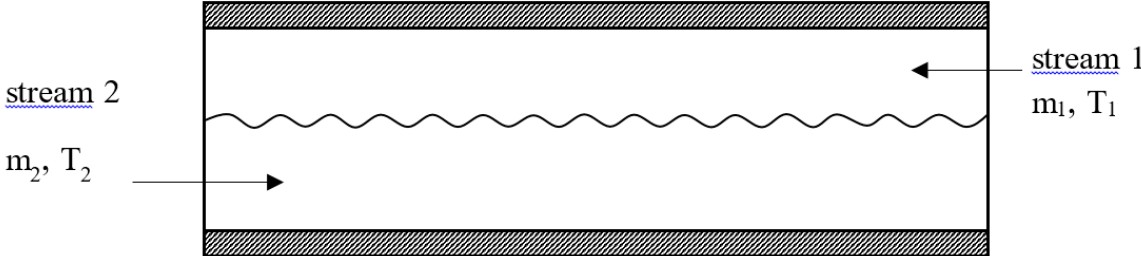

**Figure 3.** Enthalpy exchanger [21].

Typical air temperature and humidity ratio of this configuration extracted from [15] is given in Table 2, where the figures on the left of the component's name represent the inlet values and those on the right represent the outlet values.

**Table 2.** Typical (design) temperature and humidity ratio at each of the component's inlet and outlet, extracted from [15] whose configuration is shown in Figure 2. EE: Energy Exchanger, DW: Desiccant wheel, ERW: Energy recovery wheel, DEC: Direct evaporative cooler, RG: Regenerator. (The green area represents the process on the supply side whilst the blue area represents the process on the exhaust side with the arrows indicating the flow directions.).

| °C | | 32 | | 30 | | 60 | | 30 | | 25 | | Temperature |
|---|---|---|---|---|---|---|---|---|---|---|---|---|
| | | → | EE | → | DW | → | ERW | → | DEC | → | | Supply side |
| g/kg | | 18 | | 13.5 | | 5.5 | | 5.5 | | 9.5 | | Humidity ratio |
| °C | 25 | | 29 | 27 | | 25 | | 50 | | 100 | | 60 | Temperature |
| | → | EE | ↓ | → | DEC | → | ERW | → | RG | → | DW | → | Exhaust side |
| g/kg | 10.5 | | 15 | | | 18 | | 18 | | 18 | | 26 | Humidity ratio |

## 3. Overview of Climates of Brisbane, Townsville and Darwin

Brisbane (and Queensland in general) has a climate with "relatively high temperatures and evenly distributed precipitation throughout the year" (i.e., subtype Cfa—humid subtropical according to Köppen Climate classification) [22]. Townsville is in the north eastern coast of Queensland; it has a tropical climate with a hot and humid summer but with relatively low rainfall compared with other tropical cities [23]. Darwin has a tropical wet and dry or savannah climate (Aw in the Köppen–Geiger climate classification) [24,25].

Figure 4 shows the plot of the humidity ratio against the dry bulb temperature of air in Brisbane, Townsville and Darwin during the cooling months. As shown, the number of points (hours) where humidity ratios are below ASHRAE's [26] upper recommended value of 12 g/kg within 20–26 °C range in the three cities varies. Brisbane has the highest number of hours, followed by Townsville whilst numbers for Darwin are very low. The significance of this observation lies in the fact that these numbers reflect the duration during which the natural ventilation may be sufficient to provide the thermal comfort of the occupants. The higher this number the shorter the time when an air-conditioned system is required to provide thermal comfort.

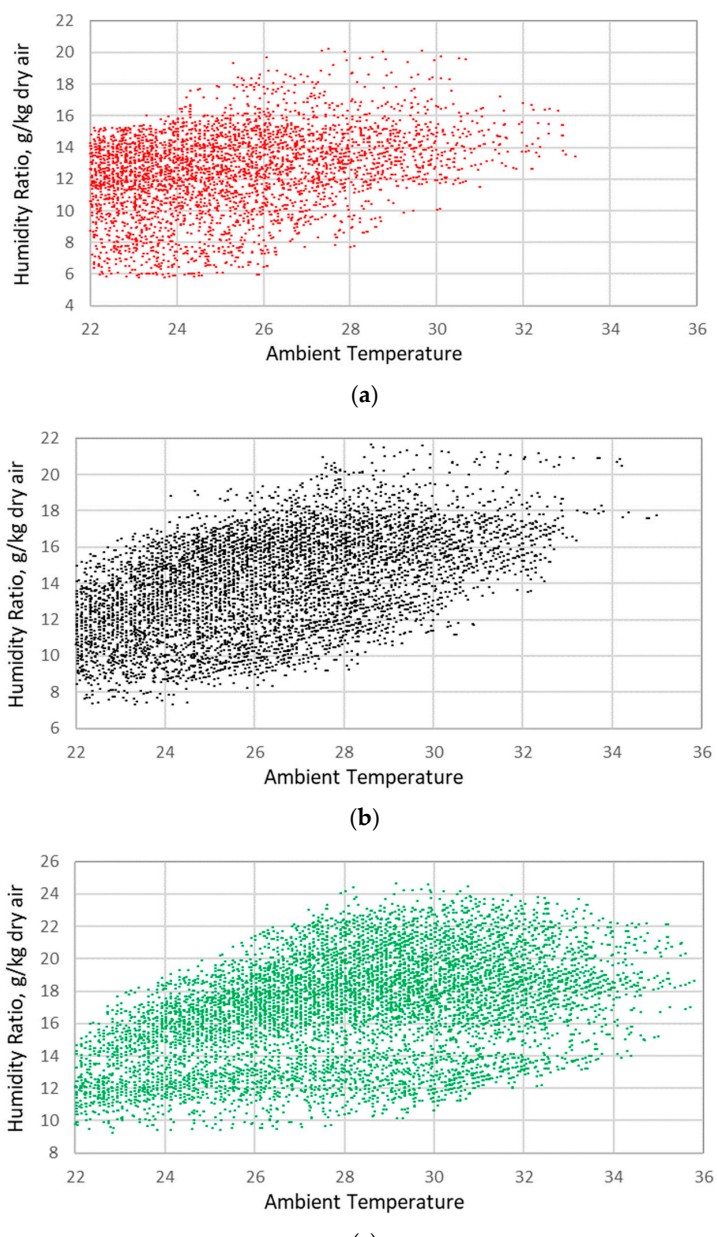

**Figure 4.** Air Humidity Ratio vs. Ambient Temperature for cooling months for (**a**) Brisbane; (**b**) Townsville; (**c**) Darwin.

Another relevant quantity that emerges from Figure 4 is the number of points (hours) within the 26–36 °C range in which the humidity ratio is below 8 g/kg, appearing in the first configuration discussed in [15], presented in Table 1 and implemented in the control strategy discussed in [8]. That is, only very few points (hours) in Brisbane, fewer in Townsville and none in Darwin of the outside air

humidity ratio below 8 g/kg mentioned in [8] that enable the control strategy to operate the system that would have saved the energy and increased the COP.

The above observation clearly indicates that for the three cities, the challenge to the SDEAC system to satisfy both the temperature and humidity levels for occupant comfort is obvious, with Darwin climate posing the biggest challenge, in particular during the build-up period [24].

## 4. Methodology

To evaluate the performance of the above configurations, TRNSYS [27] models of the cooling systems installed in houses located in the cities of Brisbane, Townsville, Darwin were developed based on the models of the following components detailed in [21]: enthalpy exchanger, desiccant wheel, energy recovery wheel, evaporative cooler and regenerator. In addition to these components, a supply fan, air mixers to model the mixing of return air and the fresh air that are conditioned before being supplied to the room form part of the system modelling. In the model, the system is installed in a three bed-room house whose specifications and thermal zoning was detailed and reported earlier in [18]. Figure 5 shows the interconnections between the components in the TRNSYS model.

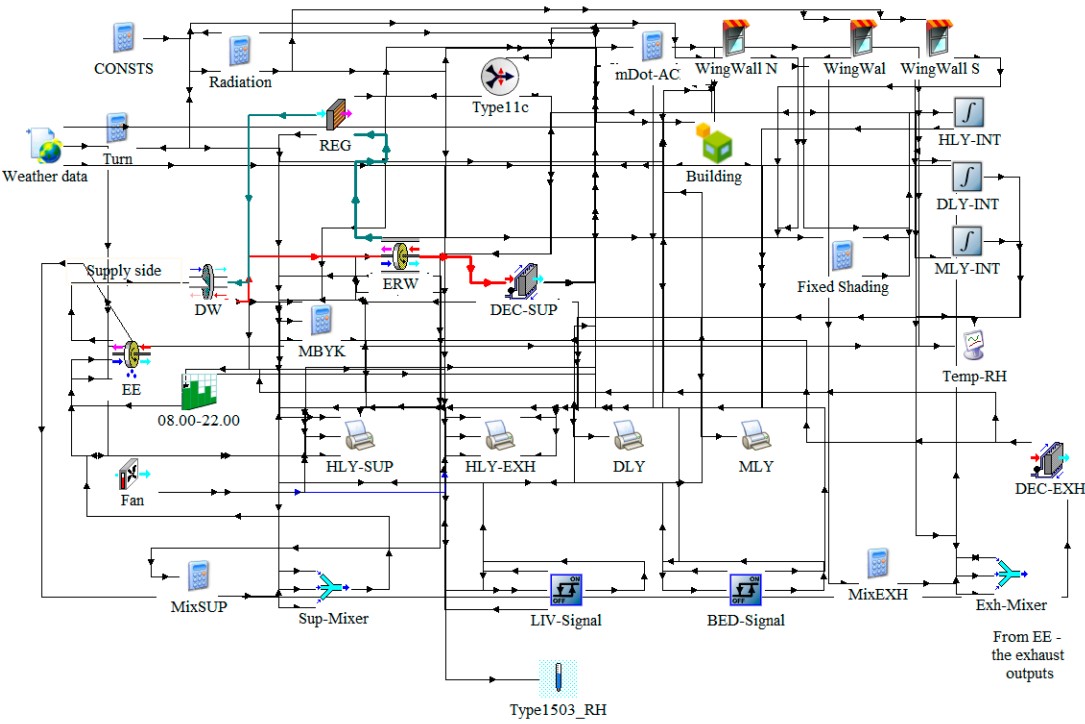

**Figure 5.** TRNSYS model of the SDEAC system.

The three bedrooms were conditioned from 23:00 to 07:00 and the combined living and kitchen room was conditioned from 07:00 to 23:00. The indoor temperature of both bed and living rooms was set to 26 °C with temperature dead band of ±0.5 K. While the SDEAC system has evaporative cooler as one of its components, the system is treated equal to the normal refrigerative system in the model in terms of its thermal comfort delivery capability. That is, it is expected to cool and dehumidify the air and not rely on the high velocity air to deliver the cooling effect. As such, the desired room humidity level is set according to the ASHRAE/Fanger comfort chart [24] adapted for Australian homes in which the acceptable indoor design temperature range of 22.5–26 °C with dew point at or below 13 °C [26]. A simple ON/OFF control strategy with ±0.5 K deadband was applied to control the room temperature.

Since the outside air conditions in the three cities are different, it is expected that the system will respond accordingly; i.e., it is expected that the system operating in the least challenging ambient

conditions—in this case Brisbane—will perform better in terms of attainment of the desired temperature and humidity ratio of the conditioned space.

Once the minimum specifications that enable the delivery of the desired temperature and humidity ratio in the least challenging environment have been established, the specification for the system operating in warmer and more humid cities—in this case Townsville and Darwin—are modified until they satisfy the thermal conditions of the conditioned spaces. The initial inputs to the TRNSYS models which satisfy the "baseline" Brisbane conditions are listed in Table 3.

**Table 3.** Main inputs to the TRNSYS models.

| **Room Temperature Set Point** | **26 °C (±0.5)** |
| --- | --- |
| Conditioned spaces | Living and bedrooms |
| Room air mass flowrate | 600 kg/h (baseline–Brisbane) |
| Room conditioning schedule | Living: 07.00–23.00 |
| | Bedroom: 23.00–07.00 |
| Number of occupants | 5 |
| Other heat gain sources | 1 computer, kitchen appliances, television set |
| Infiltration | 0.5 ACH (air change per hour) |
| Regeneration temperature set point | 100 °C |
| Regeneration heat input | 8.2 kW |
| Maximum outlet temperature of ERW | 30 °C |
| Desiccant wheel | Power consumption 0.2 kW |
| Direct evaporative cooler | Power consumption 0.1 kW |
| Energy recovery wheel | Power consumption 0.1 kW |
| Fan power | Power consumption 0.1 kW |
| Range of flow rates of the SDEAC components on the supply and exhaust sides | 600–1000 kg/h (see also accompanying text) |
| Weather data used | Brisbane, Townsville, Darwin |
| Range of latent effectiveness | 0.3–0.7 |
| Range of sensible effectiveness | 0.6–0.8 |

## 5. Results of the Simulation

### 5.1. Rooms Temperature Profiles

Figure 6 shows the air temperatures in the conditioned spaces in the three cities. The mass flow rate of supply air for these figures is set to 600 kg/h, i.e., the baseline value for Brisbane.

It is worth noting that given the same capacity (i.e., the same mass flow rate of supply air—600 kg/h), the cooling system is able to provide cooling relatively smoothly in Brisbane and fails for relatively insignificant hours in Townville and many hours in Darwin. The first two cities also experience some heating periods where cooling is not required, with heating period in Brisbane longer than in Townsville. On the other hand, Darwin always requires cooling (and dehumidification) all year round. In Darwin, the cooling system is able to maintain the room temperature within the set point during the "winter" months of June–July, coinciding with relatively low temperature and humidity ratio.

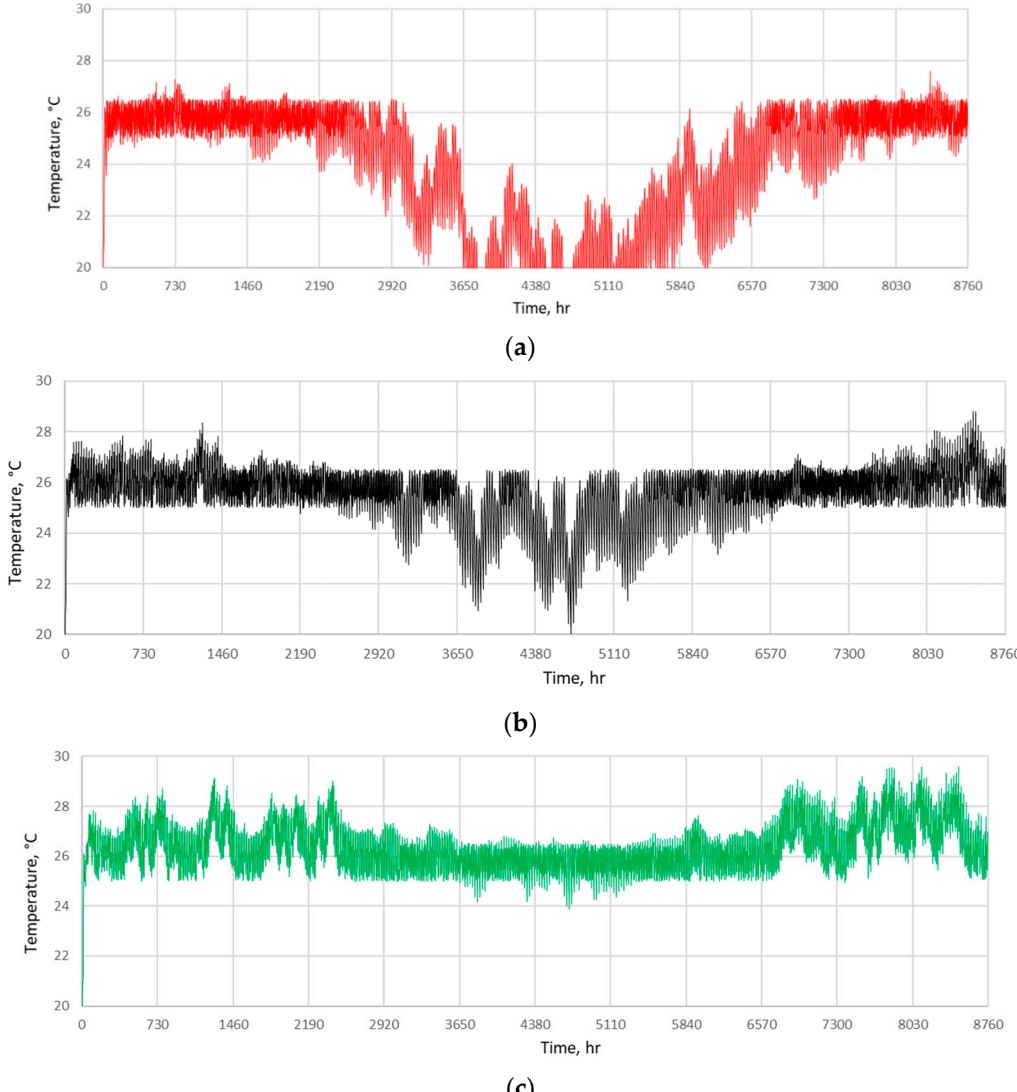

**Figure 6.** Room Temperature profiles for the three cities for the same mass flow rate of supply air, i.e., 600 kg/h. (**a**) Brisbane; (**b**) Townsville; (**c**) Darwin.

*5.2. Cooling and Dehumidification Characteristics of the Enthalpy Exchanger*

To further lower the room temperatures for Townsville and Darwin so that most of the room temperatures during the cooling periods fall within the comfort zone, the mass flow rate of supply has to increase to 1000 kg/h for Townsville and 900 kg/h for Darwin. With these values set, the cooling and dehumidification capabilities of the enthalpy exchanger were examined. In particular, the characteristics of the following parameters were studied: Latent effectiveness and sensible effectiveness. Referring to Figure 3, these two effectiveness parameters are defined as the effectiveness of the device in transferring latent and sensible energy from one flow stream to another, respectively, with respect to the potential based on the two streams at their inlet [21]. The thermal comfort zone in this section is based on the ASHRAE/Fanger comfort chart [25] for summer. In that standard, the right boundary is a straight line which passes through two points, i.e., (27.4 °C, 8 g/kg and 27 °C, 12 g/kg). The upper boundary is the 12 g/kg humidity ratio line.

5.2.1. Effect of Latent Effectiveness

Figures 7–9 show the plots of the room temperature vs. humidity ratio profiles for the three cities. For Brisbane—the city with the least humid climate—the system was able to produce the comfort

conditions for approximately 92% to 94% of the time when the latent effectiveness was increased from 0.3 to 0.7 for a constant sensitive effectiveness of 0.8. In other words, in the case of Brisbane with less humid climate, the impact of effectiveness was marginal.

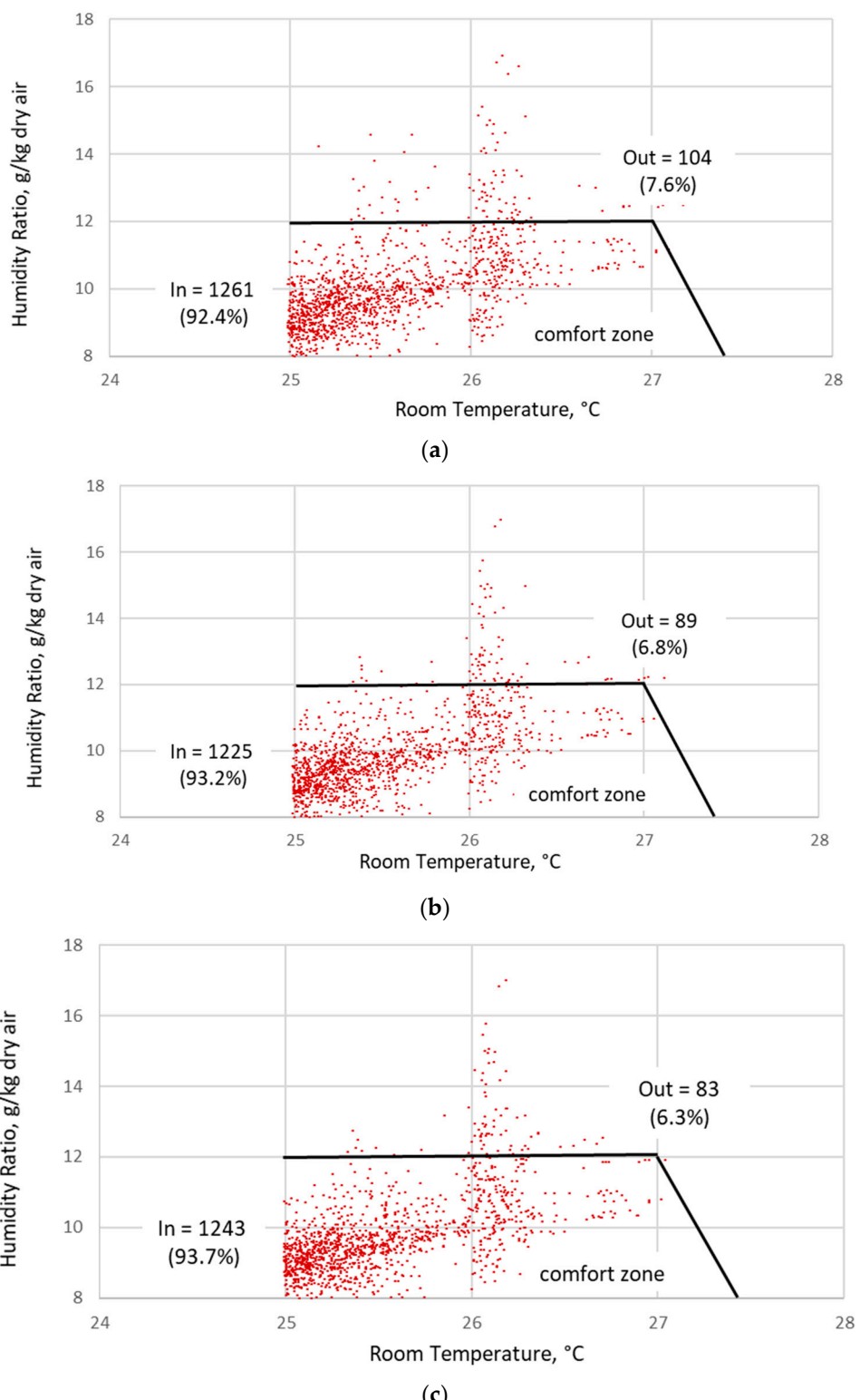

**Figure 7.** Effect of latent effectiveness on the humidity level, Brisbane, m = 600 kg/h. (**a**) EE latent effectiveness = 0.3; (**b**) EE latent effectiveness = 0.5; (**c**) EE latent effectiveness = 0.7.

For Brisbane, to keep the room temperature mostly within the comfort zone up to 26.5 °C, the mass flow rate of 600 kg/h was used. Increasing the latent effectiveness from 0.3 to 0.5 increased the number of points that fall within the comfort zone. However, further increase of the latent effectiveness to 0.7 did not make significant difference.

To attain the same effect in Townsville (i.e., to maintain the temperature within 26.5 °C), the mass flow rate of 1000 kg/h was used without affecting the humidity level. Therefore, it was decided to vary the latent effectiveness at the mass flow rate of 1000 kg/h. As shown, there was a pull of humidity ratio down towards the comfort zone when the latent effectiveness was increased from 0.3 to 0.5 and 0.7 with very insignificant impact. The majority of points being pushed still reside within the 12–14 g/kg boundary. This can be explained as follows. The increase of mass flow rate from 600 to 1000 kg/h to maintain the temperature below 26.5 °C resulted in the increase of humidity introduced by the evaporative cooler to the room. In other words, the impact of improvement in the latent effectiveness was neutralised by the evaporative cooling.

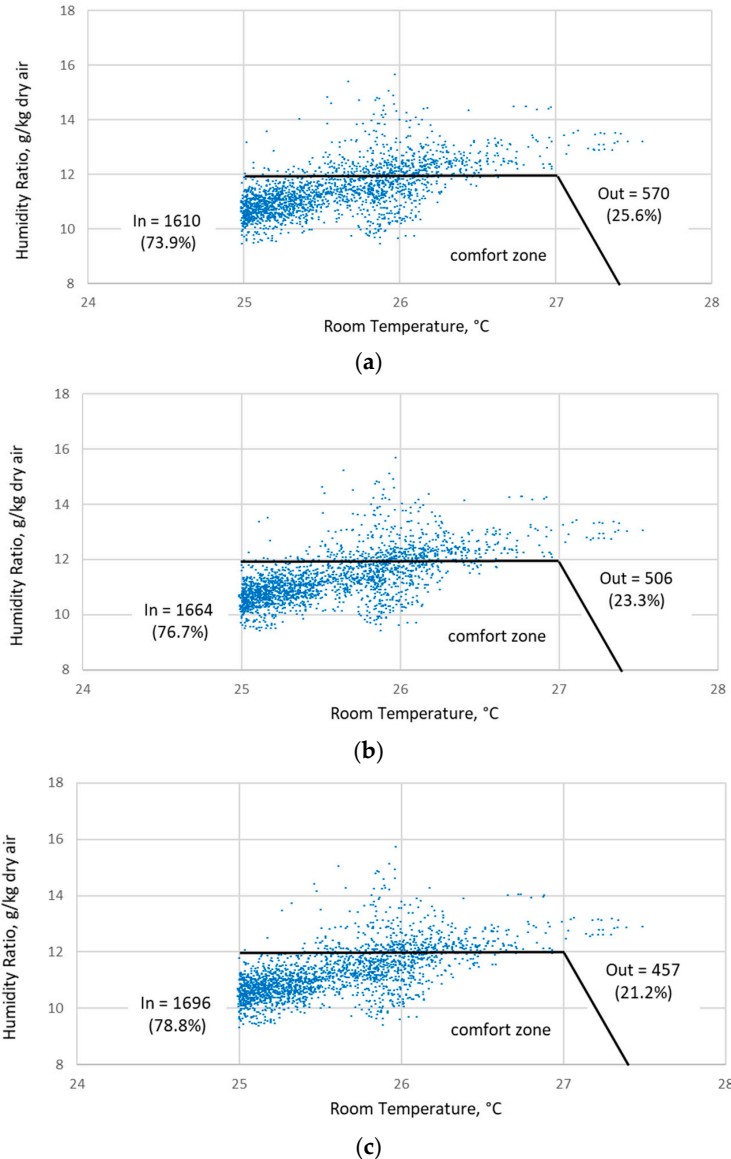

**Figure 8.** Effect of latent effectiveness on the humidity level, Townsville, m = 1000 kg/h. (**a**) EE latent effectiveness = 0.3; (**b**) EE latent effectiveness = 0.5; (**c**) EE latent effectiveness = 0.7.

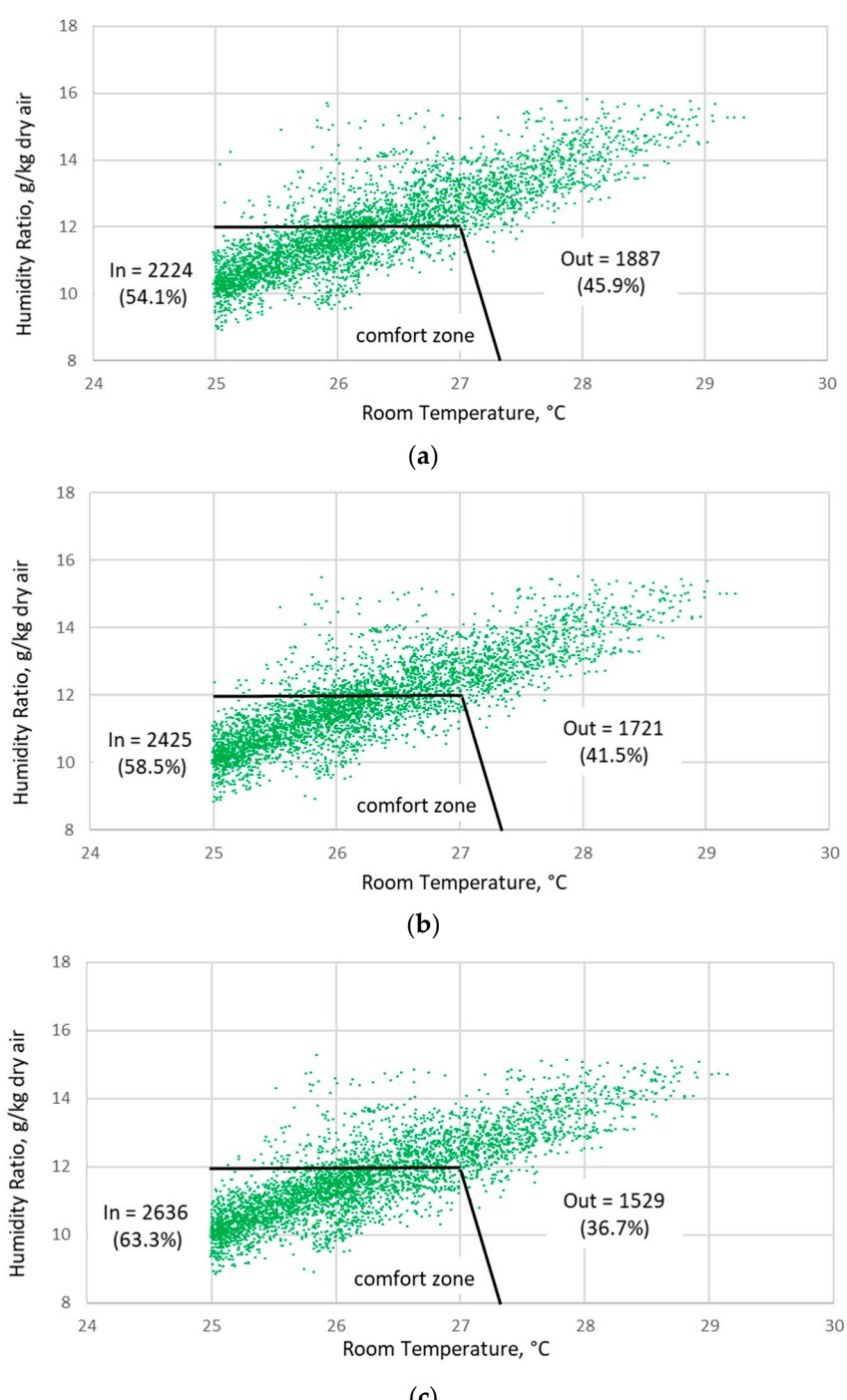

**Figure 9.** Effect of latent effectiveness on the humidity level, Darwin, m = 900 kg/h. Sensible effectiveness of 0.6. (**a**) EE latent effectiveness = 0.3; (**b**) EE latent effectiveness = 0.5; (**c**) EE latent effectiveness = 0.7.

For the Townsville climate, the system was able to produce comfort conditions within 74% to 79% of the time when the latent effectiveness was increased from 0.3 to 0.7. As shown, an increase of latent effectiveness from 0.3 to 0.7 resulted only in marginal improvement of comfort conditions (i.e., increase of number of hours in which the humidity level (and temperature) fall within the comfort zone).

The situation for Darwin was similar to that of Townsville. The mass flow rate of 900 kg/h improved the condition by decreasing the maximum temperature to about 29.5 °C while the humidity ratio for some points were pulled down. However, further increase in mass flowrate (1200–1500 kg/h) resulted in marginal cooling improvement but significant increase in the humidity ratio of air; the cause

of which was explained previously. Therefore, it was concluded that the optimum mass flow rate for Darwin for cooling purposes is 900 kg/h. This value was then used to show the impact of varying the values of the latent effectiveness of the enthalpy exchanger on the humidity ratio of room air. As shown, for the Darwin climate, the system could only create the conditions where about 54% to 63% of the points are within the comfort zone and the remaining are outside the comfort zone.

Table 4. shows the summary of the impacts of the latent effectiveness on the room thermal comfort for the three cities.

**Table 4.** The impact of the latent effectiveness on the room thermal comfort for the cities of Brisbane, Townsville and Darwin. LE: Latent Effectiveness; IN: Number of hours air conditions are within thermal comfort zone; OUT: Number of hours air conditions are outside thermal comfort zone; Total: Total hours of system operation per year.

| City | LE | IN | % | OUT | % | Total |
|------|-----|------|------|------|------|-------|
| | 0.3 | 1261 | 92.4 | 104 | 7.6 | 1365 |
| Brisbane | 0.5 | 1225 | 93.2 | 89 | 6.8 | 1314 |
| | 0.7 | 1243 | 93.7 | 83 | 6.3 | 1326 |
| | 0.3 | 1610 | 73.9 | 570 | 25.6 | 2180 |
| Townsville | 0.5 | 1664 | 76.7 | 506 | 23.3 | 2170 |
| | 0.7 | 1696 | 78.8 | 457 | 21.2 | 2153 |
| | 0.3 | 2224 | 54.1 | 1887 | 45.9 | 4111 |
| Darwin | 0.5 | 2425 | 58.5 | 1721 | 41.5 | 4146 |
| | 0.7 | 2636 | 63.3 | 1529 | 36.7 | 4165 |

### 5.2.2. Effect of Sensible Effectiveness

The following graphs in Figure 10 show how the sensible effectiveness affects the overall performance of the system, in particular the temperature of the conditioned room for Darwin. It can be seen that the sensible effectiveness of the enthalpy exchanger does not have significant impact on the cooling capability of the whole system.

The use of an enthalpy exchanger to improve the dehumidification capability of the SDEAC system was shown to have mixed impact, depending on location of system installation. For the climate of Brisbane, the system overall thermal comfort capability delivery was quite impressive; around 93% of the time when the system operates, room thermal comfort is satisfied. For the more humid city of Townsville the SDEAC system with enthalpy exchanger could satisfy occupants' thermal comfort for 74% to 79% of the system operating hours. For the hot and humid city of Darwin, this percentage dropped down to 54% to 63%. In other words, for Darwin, the SDEAC system with enthalpy exchanger did not seem to be effective to bring the thermal comfort to the conditioned space. In a previous study [18] it was found that admission of some of return air to the supply side can reduce some energy demand. However, in the exhaust side, this results in increased regeneration temperature. At all three cities studied, the impact of both latent and sensible effectiveness of the enthalpy exchanger was marginal.

These findings need to be contrasted with the conclusions of recent numerical models [6–8] which modelled a SDEAC system with the configuration shown in Figure 1 with less dehumidification capability because it does not have the enthalpy exchanger component which precools and pre-dehumidifies the air to enable it to cope with more humid climates such as those of Townsville and Darwin. In a numerical study, Dezfouli [5] presented the results of the simulation using the same configuration of Figure 1 for the hot and humid weather of Kuala Lumpur, Malaysia. For a fair comparative analysis, the ambient temperature and humidity ratio of both Darwin (which represents the warmest and most humid city under investigation) and Kuala Lumpur (which was the subject of the analysis in [5]) are shown in Figures 11 and 12, respectively. As seen, for most of the cooling months in Darwin, the ambient temperature profiles of both cities are similar. The values of ambient humidity

ratio for Darwin during the cooling season are slightly higher with the annual hourly mean of 18.5 g/kg (hourly minimum: 13.1 and maximum: 24.2 g/kg) whilst for Kuala Lumpur, the annual hourly mean is 17.7 g/kg (hourly minimum: 13.0, hourly maximum: 23.4 g/kg).

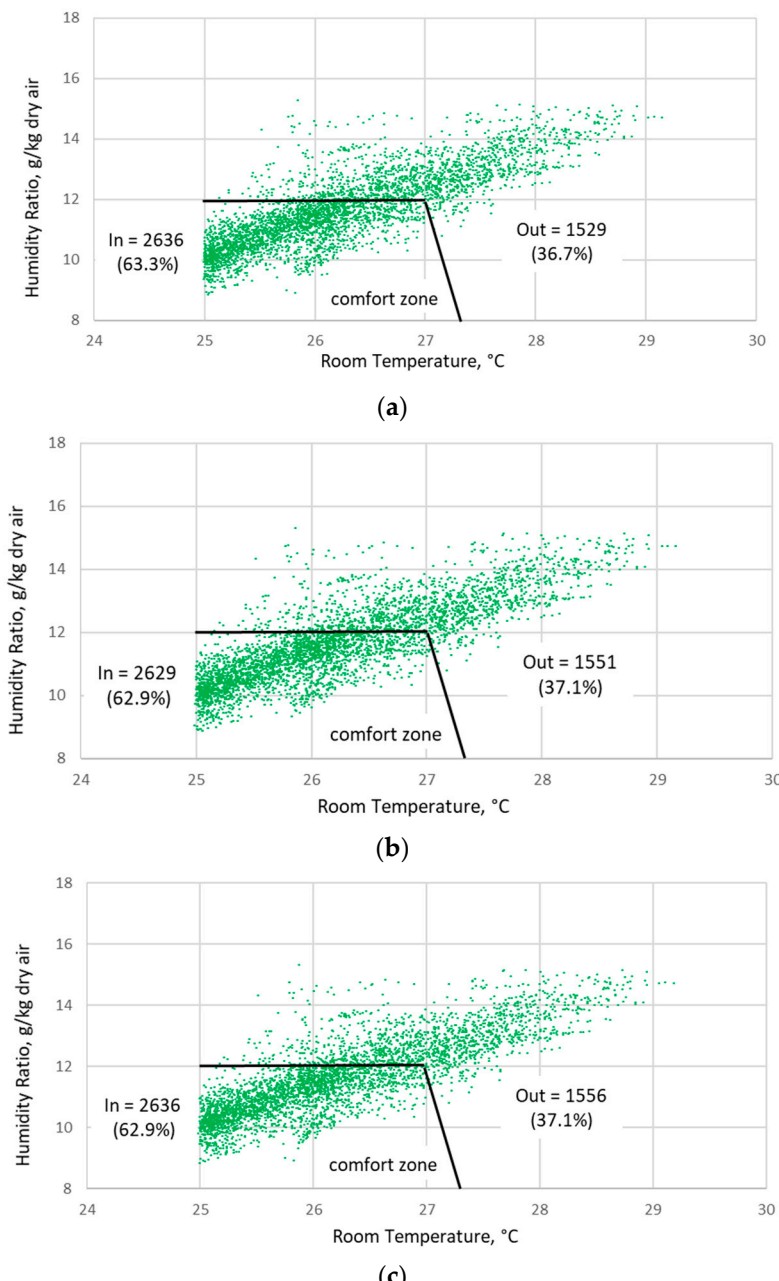

**Figure 10.** Effect of sensible effectiveness of the enthalpy exchanger, Darwin. m = 900 kg/h, latent effectiveness = 0.7. (**a**) EE sensible effectiveness = 0.6; (**b**) EE sensible effectiveness = 0.7; (**c**) EE sensible effectiveness = 0.8.

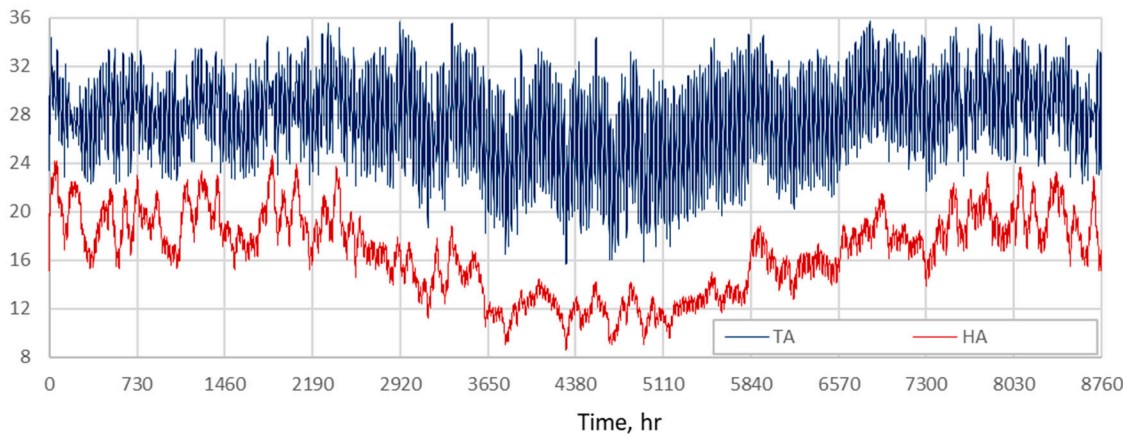

**Figure 11.** Ambient air temperature and humidity ratio of Darwin.

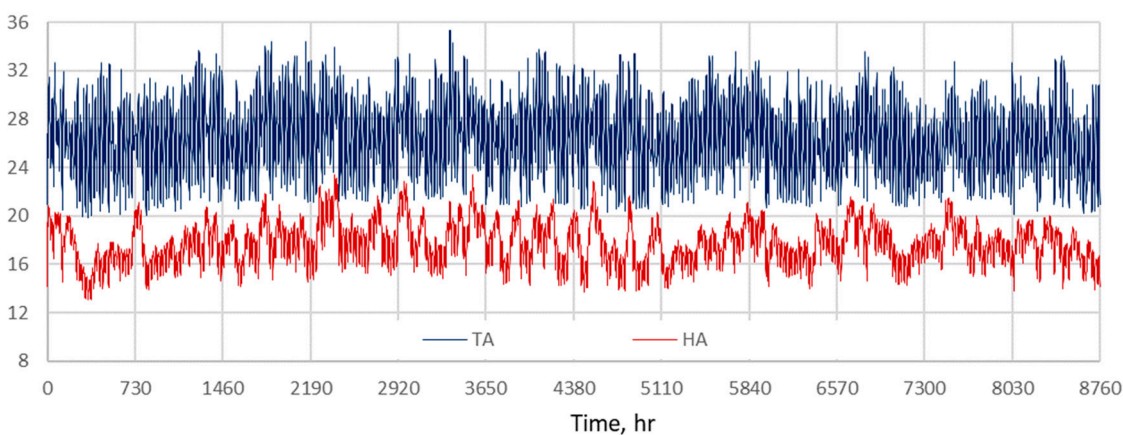

**Figure 12.** Ambient air temperature and humidity ratio of Kuala Lumpur.

Dezfouli et al.'s study [5] shows that the SDEAC system using the first configuration (Figure 1) delivers the humidity ratio of supply air which always falls outside the comfort zone (i.e., above 12 g/kg) all year round, and the resulting humidity values of the conditioned space are even higher (i.e., around 15 g/kg) all year round for both the ventilation and recirculation mode. These findings are indirectly in agreement with the results obtained from the current study (which uses the second configuration—Figure 2 with slightly higher dehumidifying capability), and with an earlier study [18]. For Kuala Lumpur, Dezfouli [5] found that only through the use of a two-stage SDEAC system did the values of resulting humidity ratio of the conditioned space fall within the comfort zone.

### 5.3. The Coefficient of Performance

In the previous Sections 5.1 and 5.2, the first key performance indicator, namely the ability to provide thermal comfort (cooling and dehumidification) to the conditioned space was presented. Following up on this indicator, which has shown to have mixed outcomes, this section presents the simulation results for the COP based on the values of the parameters used in the previous sections (see Table 3). In addition to the specifications given in Table 4, the values in Table 5 were used to calculate the COP of the cooling systems based on the results presented in Sections 5.1 and 5.2. These values reflect the specifications which give the best results in terms of thermal comfort delivery (i.e., cooling and dehumidification) for the three cities.

**Table 5.** The values of mass flow rate, latent and sensible effectiveness used to calculate the electrical and thermal COP.

| Location | $\dot{m}$ (kg/h) | Latent Effectiveness | Sensible Effectiveness |
|---|---|---|---|
| Brisbane | 600 | 0.7 | 0.8 |
| Townsville | 1000 | 0.7 | 0.8 |
| Darwin | 900 | 0.7 | 0.8 |

In this analysis, two types of coefficient of performance are presented, namely, electrical COP and thermal COP. Electrical COP is an indication of how much electrical energy the system consumes (EC) for a given conditioning (cooling and dehumidifying) load (L) and defined as:

$$COP_{EL} = L/EC$$

This indicator is normally used to compare the cooling system eligibility as an alternative to the conventional refrigerative air conditioning systems.

The thermal COP, defined as the ratio of the conditioning load in kW (L) to the thermal energy required for regeneration in kW (HR):

$$COP_{TH} = L/h$$

The electrical and thermal COP of the systems operating in the three cities are shown in Table 6. The table also includes electrical energy consumption of the system based on the parasitic power of the components listed in Table 3. As seen, the system performs better where the conditioning load is less challenging. The hotter and more humid the location where the system serves, the less efficient the system. The systems operating in Townsville and Darwin worked harder than the one operating in Brisbane. This observation is in line with the work of [5,19] but in very much contrast with the work of [6–8]. In [19], the COP is expressed as a function of ambient humidity ratio where increase in the humidity ratio decreases the COP. The same conclusion can be found in [5] for both single-stage and two-stage systems.

**Table 6.** Electrical energy consumption, thermal and thermal COP of the systems operating in Brisbane, Townsville and Darwin.

| City | EC (kWh) | $COP_{EL}$ | $COP_{TH}$ |
|---|---|---|---|
| Brisbane | 741 | 4.8 | 0.36 |
| Townsville | 1846 | 3.8 | 0.28 |
| Darwin | 3678 | 2.9 | 0.21 |

It is worth noting that the electrical COP depends very much on the actual electricity consumption of the components of the system and therefore depends on the actual manufacture specification. The main reason why the SDEAC system electrical COP is low, i.e., practically comparable with the conventional refrigerative system, is the presence of additional electrical consuming components (i.e., enthalpy exchanger, desiccant wheel and energy recovery wheel) to bring back the humidity to an acceptable level which is elevated during the evaporative cooling process just before the air enters the conditioned space. It is also worth noting that the thermal COP of this type of system is quite low and therefore should only be considered when the heat source for generation is cheap. While solar energy can be a potential source, such a huge demand for regeneration heat (e.g., in this case study 8.2 kW is required for regeneration) requires a very careful design to ensure it is worth the switch to such a system. According to [28], desiccant wheel is the component with the greatest percentage of total exergy destruction of a desiccant system; therefore, an improvement effort such as [1] needs to be prioritised.

The two-stage system was found to perform better in terms of both improved COP and improved satisfaction of conditioning load [5,19]. However, the two-stage system required additional components which added to the initial costs and increased space requirements.

## 6. Conclusions and Recommendations

Based on the results of the SDAEC system model and subsequent discussion the following conclusions can be drawn.

1.  The SDEAC system with enthalpy exchanger performs better than that without enthalpy exchanger in terms of dehumidification, and the impact depends on the climate where the system operates. Specifically, the following findings can be reported:

    a.  For Brisbane with less humid climate, the system can bring thermal comfort for around 93% to 94% of the operating hours.
    b.  For the more humid city of Townsville, the system satisfies occupants' thermal comfort for 74% to 79% of the operating hours.
    c.  For the hot and humid city of Darwin, the system thermal comfort capability drops to around 54% to 63% of the operating hours.
    d.  For all the three cities, the impacts of latent and sensible effectiveness of the enthalpy exchanger are marginal.

2.  Cooling and humidification of the direct evaporative cooler results in the existence of the optimum mass flow rate of the air supply. In a very humid location like Darwin, increasing flow rate further from the optimum results in increase in the humidity ratio of conditioned space.
3.  The electrical and thermal COP of the system depends on the system conditioning load, i.e., the higher the conditioning load, the lower the COP. The values of electrical COP for the systems operating in Brisbane, Townsville and Darwin were 4.8, 3.8 and 2.9, respectively. Likewise, the values of the thermal COP were 0.36, 0.28 and 0.21, respectively.
4.  Like previous investigations, it can be concluded that the main drawbacks of a SDEAC system are: (1) the existence of main components with conflicting thermal capabilities, i.e., desiccant wheel and evaporative cooler, and (2) the requirement of significant regeneration heat, and consequently (3) the extra components that are required to deliver the cooling and dehumidification.

This study has presented a straightforward and objective method of assessing the thermal performance of the SDEAC system, i.e., direct evaluation of its capability to deliver thermal comfort, namely, acceptable air temperature and humidity levels in the conditioned space. While performance evaluation of each of the SDEAC components is important, especially for optimisation of component thermal performance, direct evaluation of the system capability to deliver acceptable thermal comfort condition is always necessary, as this is the ultimate goal of any air conditioning system.

**Author Contributions:** R.N. initiated the project and developed the initial TRNSYS model. E.H. provided assistance in the detailed development of the TRNSYS model and provided inputs and critical analysis based on the model outcomes. S.J. provided general inputs and comments on the manuscript.

**Funding:** The funding for this project was provided by the Central Queensland University.

**Conflicts of Interest:** The authors declare no conflict of interest.

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
