# Peer review of "Dehumidification Potential of a Solid Desiccant Based Evaporative Cooling System with an Enthalpy Exchanger Operating in Subtropical and Tropical Climates"

_energies, doi:10.3390/en12142704_

Round 1

Reviewer 1 Report

The subject of solid desiccant based evaporative cooling has been extensively studied since the 1980's. Notedly by the Solar Energy Research Institute (Presently NREL) of the United States. I suggest the authors review some of these past references. Other comments are given below:

In Figure 1 ERW is a sensible heat exchanger, in the past studies efficiency of ERW is important to the system efficiency.

In Figure 2, return air from the conditioned space seems to split into two streams. The proportion of the split is not addressed.

Figure numbers have been misplaced, Figure 1-Enthalpy Exchanger comes after Figure 2.

Figure 1-Enthalpy Exchanger is decribed as has permeable membrane allow mass transfer between two streams. Please note that mixing of two streams is not advantages. Please look upon the design principles of commercial enthalpy exchangers.

Figure 3-TRNSYS model is difficult to read.

Please state the type of EE applied in this analysis.

The selection of EE latent effectiveness is hypothetical. For a small system normally cross flow type is used. Unless the face velocity is low the latent effectiveness is not high.

In Figure 5 and Figure 6, EE latent effectiveness appears to have minimum impact on the room humidity ratio.

The results presented in Figure 7, even with sensible effectiveness of 0.6 room temperature or humidtity cannot be maintained at comfort level,  please explained.

The authors state in the conclusion of electrical COP and thermal COP, please comment on the total energy consumed by the systems, especially the parasitic power of thermal energy usage.

Author Response

Response to Reviewers’ Comments

The authors would like to thank all the reviewers whose inputs and constructive comments have helped to improve the manuscript further. The following are our responses to the reviewer’s comments.

Reviewer 1

Comments and Suggestions for Authors

The subject of solid desiccant based evaporative cooling has been extensively studied since the 1980s. Notedly by the Solar Energy Research Institute (Presently NREL) of the United States. I suggest the authors review some of these past references. Other comments are given below:

Additional references of NREL work reviewed and incorporated into the paper

·         Pesaran, A.A. and Mills, A.F., Moisture transport in silica gel packed beds--I. Theoretical study. International Journal of Heat and Mass Transfer, 1987. 30(6): p. 1037-1049.

·         Pesaran, A.A., Penny, R., Czanderna, A.W., Desiccant Cooling: State-of-the-art assessment. 1999.

In Figure 1 ERW is a sensible heat exchanger, in the past studies efficiency of ERW is important to the system efficiency.

Agrees with this statement. Paper does not have any contradicting argument.

In Figure 2, return air from the conditioned space seems to split into two streams. The proportion of the split is not addressed.

The proportion of split is 50:50 and that is included in the revised manuscript.

Figure numbers have been misplaced, Figure 1-Enthalpy Exchanger comes after Figure 2.

Figure 3 Enthalpy Exchanger is placed after Figure 2.

Figure 1-Enthalpy Exchanger is described as has permeable membrane allow mass transfer between two streams. Please note that the mixing of two streams is not advantages. Please look upon the design principles of commercial enthalpy exchangers.

According to Henning [9], this normal SDEAC without EE configuration (shown in Figure 1) is suitable for the temperate climate only. So as an alternative, EE is introduced as suggested by Henning et al.  In this paper, this new system is evaluated in for different climatic conditions.

Figure 3-TRNSYS model is difficult to read.

Better figure incorporated

Please state the type of EE applied in this analysis.

It is an enthalpy wheel in which the values of both latent and sensible effectiveness are set. As such, the model can accommodate the various configuration (parallel, cross, or counterflow). See also the response below.

The selection of EE latent effectiveness is hypothetical. For a small system normally, cross flow type is used. Unless the face velocity is low the latent effectiveness is not high. The analysis includes different values of EE latent effectiveness ( 0.3, 0.5 and 0.7).

In Figure 5 and Figure 6, EE latent effectiveness appears to have minimum impact on the room humidity ratio.

Figure 5 and 6 do not show EE latent effectiveness at all.  EE latent effectiveness is shown in Figures 7-9.  Yes. EE latent effectiveness appears to have minimum impact on the room humidity ratio.

The results presented in Figure 7, even with sensible effectiveness of 0.6 room temperature or humidity cannot be maintained at comfort level,  please explain.

This shows that the dehumidification achieved is not enough in this case and thereby comfort condition is not achieved. The discussion and conclusions in paper point to the technical limitations of this system, something that is not revealed in previous studies/investigations.

The authors state in the conclusion of electrical COP and thermal COP, please comment on the total energy consumed by the systems, especially the parasitic power of thermal energy usage.

Table 6 has been updated to include the electrical energy consumption based on the parasitic power of the components listed in Table 3.

Reviewer 2 Report

Ramadas Narayanan et al. have studied the Dehumidification Potential of a Solid Desiccantbased Evaporative Cooling System with an Enthalpy Exchanger Operating in Subtropical and Tropical Climates. In my opinion, this work is exciting, but it needs some corrections. My suggestions are as follows:

1-     It would be better for the authors to highlight the state of the art of the study in the last paragraph of the introduction. The object of study is not clear. The author should state the objectives of the current study in the last paragraph in the introduction section.

2-     It would be better for the authors to arrange the summary of each section follows the sequence.

3-     The first seven lines of introduction are too general and needed to be addressed

4-     Please improve Table.1 and Table.2.

5-     Please improve Fig.1 and Fig.2.

6-     Please unified the format of all the charts.

7-     The Comparative-quantitative results should be mentioned in the Abstract.

8-     All the parameters that have been used in this paper should be non-dimensional.

Author Response

Response to Reviewers’ Comments

The authors would like to thank all the reviewers whose inputs and constructive comments have helped to improve the manuscript further. The following are our responses to the reviewer’s comments.

Reviewer 2

Ramadas Narayanan et al. have studied the Dehumidification Potential of a Solid Desiccant based Evaporative Cooling System with an Enthalpy Exchanger Operating in Subtropical and Tropical Climates. In my opinion, this work is exciting, but it needs some corrections. My suggestions are as follows:

1-     It would be better for the authors to highlight the state of the art of the study in the last paragraph of the introduction. The object of study is not clear. The author should state the objectives of the current study in the last paragraph in the introduction section.

The introduction is updated with an additional sentence. The project investigates the effects of adding an enthalpy exchanger in the desiccant evaporative cooling system cycle for subtropical and tropical climates.

2-     It would be better for the authors to arrange the summary of each section follows the sequence.

The paper is structured with different sections to have a continuous flow of ideas as per a logical sequence.

3-     The first seven lines of introduction are too general and needed to be addressed

The introduction is changed.

4-     Please improve Table.1 and Table.2.     Please improve the Fig.1 and Fig.2. Please unified the format of all the charts.

The captions of both Tables explain quite clearly the quantities and values, and the meaning of the symbols in both tables.   The same applies to figures 1 and 2 where all symbols have been explained.

5-     The Comparative-quantitative results should be mentioned in the Abstract.

The abstract is updated by adding the sentence: It is found that the system with enthalpy exchanger performs better than that without enthalpy exchanger in terms of dehumidification, however, the impact depends on the climate where the system operates.

6-     All the parameters that have been used in this paper should be non-dimensional.

The actual parameters are used to facilitate easy understanding of for all levels of readers.

Reviewer 3 Report

The paper introduces a study on a solid desiccant  based cooling system. The paper is interesting but, in my opinion, some aspects should be improved in order to enhance the quality up to make it ready for publication.

1. Introduction should  be more exhaustive and in my opinion some further important papers for state of the art building, should be included:

-  The application of a desiccant wheel to increase the energetic performances of a transcritical cycle. Energy conversion and management, 89, 222-230, 2015.

-  Multi-stage desiccant cooling system for moderate climate. Energy Conversion and Management, 177, 77-90, (2018).

2. When authors introduce an acronym, they should write (in the extended definition) capital letters composing the acronym.

3. Authors are invited to not use abbreviations as "system's". This does not confer the appropriate formality as a scientific paper requires.

4. When you introduce the cities object of the investigation, please do it providing the nation to which they belong and the latitude and longitude coordinates.

5. Most of the citations are not visible in the text. There is "Error! Reference source non found....."

6. Tables are not easily readable. Please format them as the Energies templates requires.

7. Section 3. Authors should report data about TMY irradiations and temperatures of the cities investigated.

8. Figure 4 is not easily readable if the x-axis report time only in hours. Authors should specify the month and the related season with respect to the cities investigated.

9. Table 4. Add units of measure.

10. Nomenclature check. I am not sure that all the symbols used in the text are reported correctly in the nomenclature section. Authors should check it.

11. Please revise the English of the paper: many typos.

Author Response

Response to Reviewers’ Comments

The authors would like to thank all the reviewers whose inputs and constructive comments have helped to improve the manuscript further. The following are our responses to the reviewer’s comments.

Reviewer 3

The paper introduces a study on a solid desiccant  based cooling system. The paper is interesting but, in my opinion, some aspects should be improved in order to enhance the quality up to make it ready for publication.

1. The introduction should  be more exhaustive and in my opinion, some further important papers for state-of-the-art building, should be included:

-  Multi-stage desiccant cooling system for moderate climate. Energy Conversion and Management, 177, 77-90, (2018).

This article is included  and discussed

-  The application of a desiccant wheel to increase the energetic performances of a transcritical cycle. Energy conversion and management, 89, 222-230, 2015.

Authors believe this article is far from the scope of the current paper, so not included.

2. When authors introduce an acronym, they should write (in the extended definition) capital letters composing the acronym.

All acronyms are expanded when it is used the first time.

3. Authors are invited to not use abbreviations as "system's". This does not confer the appropriate formality as a scientific paper requires.

Abbreviations used are expanded.

4. When you introduce the cities object of the investigation, please do it providing the nation to which they belong and the latitude and longitude coordinates.

Cities included Brisbane, Townsville and Darwin are Australian cities in different climate zones.  This has been added article when the cities are mentioned the first time. This information has been added in the last sentence of the last paragraph on page 2.

5. Most of the citations are not visible in the text. There is "Error! Reference source non found....."

When it was submitted all references were visible. Editorial office modified the file before sending to reviewers. This may have caused some issues. The matter is raised with the editorial office.

6. Tables are not easily readable. Please format them as the Energies templates require.

Yes, editorial office revised the table before sending to reviewers. The matter is raised with the editorial office.

7. Section 3. Authors should report data about TMY irradiations and temperatures of the cities investigated.

The weather data files used in the model has this data. The article already has 21 pages, Authors believe that including these details will make the paper too long.  So not included.

8. Figure 4 is not easily readable if the x-axis report time only in hours. Authors should specify the month and the related season with respect to the cities investigated.

Figure 4 shows the plot of the humidity ratio against the dry bulb temperature of air for the three cities where the -axis is ambient temperature. With this plot, the reader can have an idea about the ambient conditions which need to be treated by the system. Any air conditioning system needs to bring down the humidity level so that in the end the air conditions in the room fall within the ASHRAE comfort one. The larger the number of plots being further away from the comfort one, the more work need to be performed by the system.

9. Table 4. Add units of measure.

The table has data regarding LE (Latent Effectiveness); IN( number of hours) which are just numbers and has no unit. This is already mentioned in the table title.

10. Nomenclature check. I am not sure that all the symbols used in the text are reported correctly in the nomenclature section. Authors should check it.

Checked and updated.

11. Please revise the English of the paper: many typos.

The article is revised and removed all typos.

Reviewer 4 Report

The paper presents the study of a dehumidification system including an enthalpy heat exchanger in the Brisbane,Townsville and Darwin climatic conditions. The paper is well organised and written but, in order to be sutiable for publishing, lacks scientific soundness. The authors present a numerical study in TRNSYS, but no information is given on the validation of the models for the components of the dehumidification system, thus not guaranteeing the validity of the results presented. In addition, an appendix with all the parameters used in the TRNSYS model must be added, to allow the readers to better evaluate and replicate/compare the presented results. The specifications of the enthalpy exchanger used should be given more in details.

It is also not clear what is innovative in the configuration proposed: if the aim of the author is to give guidelines for the installation of the system in different climates, results should be correlated to the different factors influencing the performance by adding a sensitivity analysis, i.e. different cooling loads, irradiation, control strategy and so on.

Author Response

Response to Reviewers’ Comments

The authors would like to thank all the reviewers whose inputs and constructive comments have helped to improve the manuscript further. The following are our responses to the reviewer’s comments.

Reviewer 4

The paper presents the study of a dehumidification system including an enthalpy heat exchanger in the Brisbane, Townsville and Darwin climatic conditions. The paper is well organised and written but, in order to be suitable for publishing, lacks scientific soundness. The authors present a numerical study in TRNSYS, but no information is given on the validation of the models for the components of the dehumidification system, thus not guaranteeing the validity of the results presented. In addition, an appendix with all the parameters used in the TRNSYS model must be added, to allow the readers to better evaluate and replicate/compare the presented results. The specifications of the enthalpy exchanger used should be given more in details.

The model of SDEAC systems is developed using TRNSYS, which is a well-established software in the HVAC industry, with all the all components validated with experimental results.  In addition, the paper is not intended to discuss the details of individual components. Our main contribution in the paper from the modelling perspective is the assembling of a number of TRNSYS component models that were used to model the entire system with an aim to investigate the effect of adding enthalpy exchanged into the system.  This assembly has enabled us to study the performance characteristics of the system subject to three distinct climatic conditions.

All the parameters of the TRNSYS model are included in Table 3.

 It is also not clear what is innovative in the configuration proposed: if the aim of the author is to give guidelines for the installation of the system in different climates, results should be correlated to the different factors influencing the performance by adding a sensitivity analysis, i.e. different cooling loads, irradiation, control strategy and so on.

The paper does not aim to provide guidelines for installation but aims to investigate the effect of adding EE in SDEAC system in different climatic conditions i.e. subtropical and tropical conditions. This investigation will help to assess the feasibility of this system under these climatic conditions. In addition, the other three reviewers agreed that this research design is appropriate,

It is stated in section 2 of the paper

According to Henning [9], this normal SDEAC without EE configuration (shown in Figure 1) is suitable for the temperate climate only. For a more humid climate, the dehumidification capacity of such a system is not adequate to enable the evaporative cooling system to provide the cooling at an acceptable room humidity level. This observation is supported by a recent numerical study for the subtropical climate of Brisbane [13] where for a significantly high number of hours the resulting humidity levels fall beyond the ASHRAE upper thermal comfort boundary. Thus, although according to [6,7,8] such a system has a relatively high electrical and thermal coefficient of performance in Darwin and Brisbane, the system may not be able to provide the comfortable humidity levels in the conditioned spaces.

So as an alternative EE is introduced as suggested by Henning et al.  In this paper, this new system is evaluated in for different climatic conditions.

Another important contribution of the paper to the research community is to show how effectively an air system can actually deliver acceptable thermal comfort conditions in the room. Previous investigations have largely attempted to study the performance of each component or to show the conditions of supply air to the conditioned room. But so far, researchers have ignored the main purpose of the installation of an air conditioning system, to provide acceptable thermal comfort. This is stated in the last paragraph of the paper

This study has presented a straightforward and objective method of assessing the thermal performance of the SDEAC system, i.e. direct evaluation of its capability to deliver the thermal comfort, namely: acceptable air temperature and humidity levels in the conditioned space. While performance evaluation of each of the SDEAC component is important - especially for optimisation of component thermal performance – direct evaluation of the system capability to deliver acceptable thermal comfort condition is always necessary, as this is the ultimate goal of any air conditioning system.   

Round 2

Reviewer 4 Report

the authors have replied to the concerns raised and therefore the paper can be published.